# Infertility screening in unmarried women: A scoping review protocol

**Sanam Borji-Navan**[1], **Nasser Mogharabian**[2*]

1 Student Research Committee, School of Nursing and Midwifery, Shahroud University of Medical Sciences, Shahroud, Iran, 2 Assistant Professor of Andrology and Urology, Sexual Health and Fertility Research Center, Shahroud University of Medical Sciences, Shahroud, Iran

* dr.mogharabian@gmail.com

## Abstract

### Objective

This scoping review aims to systematically map the landscape of infertility screening in unmarried women.

### Introduction

Infertility screening in unmarried women represents a significant and often neglected area within reproductive health. This population faces unique challenges and barriers, including social stigma, cultural norms, and limited access to care, making a comprehensive understanding of current screening practices essential.

### Inclusion criteria

Study selection will be guided by the PCCT framework (Population, Concept, Context, and study type), considering diverse study designs (quantitative, qualitative, mixed-methods, reviews) and grey literature, focusing on infertility screening in unmarried women.

### Methods

This scoping review will follow the PRISMA-ScR guidelines and utilize a 14-step framework based on Arksey and O'Malley's methodology, incorporating enhancements by Tricco and Peters. A comprehensive search strategy will be employed, using controlled vocabulary and free-text method. Databases of Web of Science (ISI), PubMed, Scopus and search engines like Google Scholar will be searched, and supplemented by forward and backward citation tracking. Inclusion/exclusion criteria will be applied iteratively. Two independent reviewers will screen titles/abstracts and full texts, resolving disagreements through consensus or a third reviewer. Data will be charted using a predefined template, and findings will be presented in tables and diagrams, accompanied by a narrative synthesis.

**Data availability statement:** No datasets were generated or analysed during the current study. All relevant data from this study will be made available upon study completion.

**Funding:** The author(s) received no specific funding for this work.

**Competing interests:** The authors have declared that no competing interests exist.

## Discussion

This scoping review will provide a comprehensive overview of the current state of knowledge regarding infertility screening in unmarried women, a significantly under-researched area. The findings will be critical for informing the development of culturally sensitive guidelines, targeted interventions, and future research to address the reproductive health needs of this underserved population.

## Introduction

Infertility, defined as the inability to achieve pregnancy after 12 months of regular unprotected sexual intercourse, is a significant global health issue affecting approximately 10–15% of couples of reproductive age [1]. While infertility encompasses both male and female factors [2], women frequently experience a disproportionate burden of social, emotional, and psychological consequences, particularly in societies where cultural norms tie reproductive capacity to marital status [3]. Within this context, unmarried women constitute a unique and often understudied population, facing distinct challenges in accessing infertility screening and care. These challenges are compounded by societal stigma, legal restrictions, and limited healthcare resources, which may delay diagnosis and exacerbate reproductive health disparities [4].

In females, the primary focus of infertility screening is on assessing ovarian reserve and uterine health. Infertility screening involves clinical assessments, hormonal assays (e.g., FSH, LH, AMH), and imaging techniques (e.g., pelvic ultrasonography) to evaluate reproductive status [5]. In specific cases, advanced procedures such as hysteroscopy or laparoscopy may be employed [6]. These measures collectively facilitate the identification of infertility etiologies and guide subsequent therapeutic interventions.

The importance of infertility screening for unmarried women is increasingly recognized, yet significant barriers persist [7]. Societal perceptions often frame fertility concerns as relevant only to married individuals, marginalizing unmarried women who seek reproductive health services [4,8]. This stigma may deter timely screening, potentially leading to poorer health outcomes due to delayed identification of treatable conditions such as tubal factor infertility or ovulatory dysfunction [9–11]. Furthermore, unmarried women may encounter structural obstacles, including lack of partner involvement, financial constraints, and restrictive policies governing access to fertility treatment [12,13].

Guidelines concerning fertility screening and counseling are limited [7]. Despite these considerations, based on our knowledge the current literature lacks a synthesis of evidence specific to infertility screening in unmarried women. This gap hinders the development of targeted guidelines and equitable healthcare policies. To address this, the present scoping review protocol outlines a structured approach to map existing research, identify knowledge gaps, and elucidate key concepts related to infertility screening in this population.

## Objectives

### Primary outcomes

1. To identify existing guidelines for female infertility screening and evaluate their applicability and specific recommendations for unmarried women.

2. To map the range and types of infertility screening methods currently used, or that could potentially be used, for unmarried women.

### Secondary outcomes

1. To synthesize the advantages and disadvantages of infertility screening specifically for unmarried women.

2. To identify knowledge gaps, areas requiring further research, and policy implications related to infertility screening in unmarried women.

## Review question

What is the current state of knowledge regarding infertility screening in unmarried women?

## Inclusion and exclusion criteria

Study eligibility was determined using the PCCT framework, detailed in Table 1.

Inclusion and exclusion criteria may be refined iteratively. Machine translation ensured language inclusivity. The search covered publications from inception to May 31, 2025.

## Methods and analysis

This scoping review, following PRISMA-P (protocol) (S1 Checklist) [14] and PRISMA-ScR (reporting) [15] guidelines, utilizes a 14-step framework. This framework, based on Arksey and O'Malley's [16] methodology and incorporating improvements by Tricco and Peters [17,18], includes: protocol development, question/objectives formulation, eligibility criteria definition, literature search (in resources, reference lists and grey literature), which will be informed by engagement with key stakeholders to refine the search strategy and identify relevant grey literature, study screen, study selection, design and perform data charting, results and flowchart presentation, and identification of research and practical implications.

### Search methods and sources (search strategy)

To ensure comprehensive literature retrieval, this review's search strategy, compliant with PRISMA-S [19], identified key concepts and keywords. This involved utilizing established thesauri (MeSH, EMTREE, ERIC), free-text method as well as employing wildcards and truncation to broaden the search terms. Database-specific search strategies were developed for

Table 1. PCCT framework.

|  | Property | Inclusion criteria | Exclusion criteria |
|---|---|---|---|
| P | **Participants** | Unmarried women (women who have never been married). | Married women. |
| C | **Concept** | Infertility Screening | Studies not related to infertility screening |
| C | **Context** | Any geographical location and setting. | – |
| T | **Types of sources** | Quantitative and Qualitative studies, Mixed-methods studies, Reviews, Grey literature (reports, guidelines, policy documents, conference abstracts and proceedings (if they provide sufficient detail), dissertations). | Editorials, letters, commentaries without original data, Duplicates |

Web of Science (Institute of Scientific Information (ISI)), PubMed, Scopus, and search engines such as Google Scholar, using Boolean operators ("OR" and "AND") appropriately (S1 File). Forward and backward citation tracking of influential articles, further enhanced the search. A detailed log of search terms and strategies will be provided in Supplementary File.

Initial search for studies will be conducted by S.B. (the principal investigator). We will not be seeking assistance from an external expert in Library and Information retrieval for the search itself. However, to ensure the robustness of our search strategy, we will consult with subject-matter experts in the field when developing and refining the keywords used.

## Study records

**Data management.** Retrieved records will be managed in HubMeta [20], with duplicates removed. Two independent reviewers will screen titles and abstracts, followed by full-text review of potentially eligible articles.

**Selection process.** Study selection will involves two phases: title/abstract screening and full-text review against pre-defined criteria. Disagreements will resolve by consensus or a third reviewer. A PRISMA flowchart (Fig 1) will detail the process. Authors will be contacted for missing data. Inter-rater reliability (Kappa) will be calculated.

**Data collection and analysis.** Two independent extractors will thoroughly review the full text of each study that meets the inclusion criteria using a standardized data charting form. This form, developed and piloted by the research team, aligns with the PCCT framework and the objectives of the scoping review.

It will systematically capture key study characteristics, including authors, publication year, study design, country, and language. Additionally, participant characteristics such as age range, subgroups, recruitment methods, sample size, and

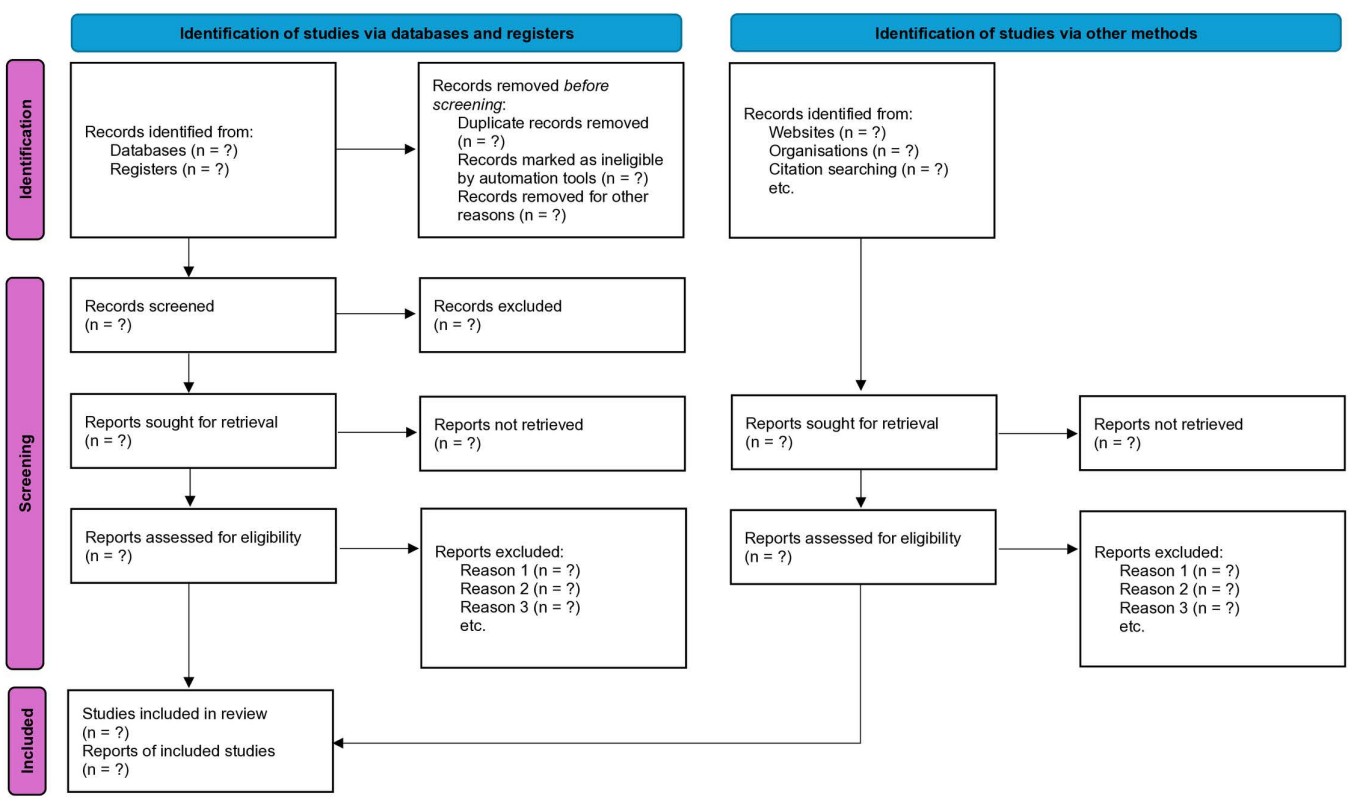

**Fig 1. PRISMA flowchart.**

contextual details will be documented. The form will also address infertility screening details, including the types of methods discussed, motivations for screening. Contextual details will cover the study setting (clinical, community, etc.) and relevant geographical or cultural influences.

The process of data extraction will be iterative, with the form being reviewed and refined based on initial extractions to ensure comprehensive data capture. Any discrepancies between the two extractors will be resolved through discussion, and if necessary, a third reviewer will be consulted. The extracted data will be synthesized into tables and diagrams, accompanied by a narrative summary that highlights key themes, gaps, and the overall scope of research in this area.

S.B. will be contacted for any essential missing information to ensure a complete data synthesis. A rigorous quality assurance process will be implemented to address discrepancies, involving consensus-building with a third researcher to guarantee the accuracy of the data. Finally, the finalized data will be systematically categorized and organized for efficient analysis.

### Critical appraisal

Consistent with scoping review methodology, a formal quality appraisal of included studies will not be conducted [15].

### Data synthesis and analysis

Data will be synthesized and presented in tables and diagrams, aligned with the review's objectives, and supported by a narrative summary. For qualitative and mixed-methods studies, we will employ thematic analysis.

### Ethics and dissemination

Any protocol amendments will be documented. Due to data complexity, results may be disseminated in multiple publications.

## Discussion

This scoping review protocol addresses a critical gap in the literature by outlining a systematic approach to mapping the evidence on infertility screening in unmarried women. By identifying relevant guidelines, mapping screening methods, and synthesizing their advantages and disadvantages, we anticipate that this review will identify key knowledge gaps and inform the development of targeted interventions to improve reproductive health outcomes for this underserved population.

A key strength of this protocol is its adherence to PRISMA-ScR guidelines, ensuring transparency and reproducibility.

This study will provide a critical foundation for future research, policy, and practice related to infertility screening in unmarried women.

## Supporting information

**S1 Checklist. PRISMA-P 2015 checklist.**
(DOCX)

**S1 File. Search strategy (Terms can be used in the search strategy in PubMed database).**
(DOCX)

## Acknowledgments

The authors acknowledge using an AI language model for translation, writing, and editing assistance, retaining full responsibility for the final content.

## Author contributions

**Conceptualization:** Sanam Borji-Navan, Nasser Mogharabian.

**Methodology:** Sanam Borji-Navan, Nasser Mogharabian.

**Project administration:** Sanam Borji-Navan.

**Supervision:** Nasser Mogharabian.

**Validation:** Nasser Mogharabian.

**Writing – original draft:** Sanam Borji-Navan.

**Writing – review & editing:** Sanam Borji-Navan, Nasser Mogharabian.

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
