## [Decision Letter · Decision Letter 0]

13 May 2025

We look forward to receiving your revised manuscript.

Kind regards,

Godwin Banafo Akrong, Ph.D.

Academic Editor

PLOS ONE

2. In the online submission form you indicate that your data is not available for proprietary reasons and have provided a contact point for accessing this data. Please note that your current contact point is a co-author on this manuscript. According to our Data Policy, the contact point must not be an author on the manuscript and must be an institutional contact, ideally not an individual. Please revise your data statement to a non-author institutional point of contact, such as a data access or ethics committee, and send this to us via return email. Please also include contact information for the third party organization, and please include the full citation of where the data can be found.

3. Please remove your figures from within your manuscript file, leaving only the individual TIFF/EPS image files, uploaded separately. These will be automatically included in the reviewers’ PDF.

Reviewers' comments:

Reviewer's Responses to Questions

**Comments to the Author**

1. Does the manuscript provide a valid rationale for the proposed study, with clearly identified and justified research questions?

Reviewer #1: Yes

Reviewer #2: Partly

2. Is the protocol technically sound and planned in a manner that will lead to a meaningful outcome and allow testing the stated hypotheses?

Reviewer #1: Partly

Reviewer #2: Partly

3. Is the methodology feasible and described in sufficient detail to allow the work to be replicable?

Reviewer #1: Yes

Reviewer #2: Yes

4. Have the authors described where all data underlying the findings will be made available when the study is complete?

Reviewer #1: Yes

Reviewer #2: Yes

5. Is the manuscript presented in an intelligible fashion and written in standard English?

Reviewer #1: Yes

Reviewer #2: Yes

You may also provide optional suggestions and comments to authors that they might find helpful in planning their study.

Reviewer #1: This is an interesting area and the core elements of a review protocol have been included. However I have a few comments for the authors.

• What is your definition of unmarried women? Is it never been married? Is it those who have been married before and are currently divorced/single? Is it those who are cohabiting? As this might have a bearing on your articles

• Why are you restricting to 1990-2025 period? You might need to justify

• Line 20 is the Lead author part of the independent extractors?

• Line 33 – You might need to write ISI in full for the first time

• I think you should also mention the use of wild cards and truncations to broaden the database search

• Are you also going to including the 6th optional process by Arksey and O’Malley of consultative exercise as this might provide useful information if key stakeholders are engaged in the finalization of search strategy and grey literature identification

• May you consider provision of an example of the search strategy key words table

• Take note that some words such as sub-fertility, bareness, sterility and infecundity might need to be included in the search strategy

• Who will do the search of studies? Will it be done by one person the lead author or by the independent reviewers? Will this be done separately? Are you involving any expert in Library and Information retrieval to assist with the searching?

• Line 135 Who is S.B? Is this the lead author being referred to? Who will contact him/her?

• Line 141 Joanna Briggs (JBI) have developed critical appraisal tools which you might consider to use

• Line 144 You might need to include thematic analysis for the qualitative and mixed methods studies

• Line 154 I would propose that you rephrase that since quality assessment is optional, the authors will not consider doing that process.

• Line 156 This is debatable on the usefulness of the finding for policy if the quality of studies are not assessed/ critically appraised. You might need to consider revising to include critical appraisal

• References – You might need to consider citing articles that are more current e.g 5 years since there is quite a number of literature on infertility. The methodological citations are ok.

Reviewer #2: Thank you for submitting your paper on this important and timely topic. Below are my suggestions to strengthen the manuscript:

1. Critical Discussion of Gaps and Methods:

• It would be valuable to include a critical discussion of the identified gaps in male infertility screening (e.g., limitations in current guidelines, underrepresented populations, or unmet needs in screening protocols).

• Additionally, a summary of the methods currently used for infertility screening would provide useful context for readers.

2. PRISMA Flow Diagram:

o The PRISMA diagram is currently empty, please confirm whether this is an upload error. If not, the absence of this data raises concerns about the study’s findings, as the screening process is a cornerstone of a scoping review.

o I recommend moving the PRISMA diagram from the appendices to the Methods section to improve readability and allow readers to follow the screening process seamlessly. Accompanying the diagram with a brief narrative (e.g., number of records screened, excluded, and reasons for exclusion) would further enhance clarity.

3. Methods Section Clarity and Completeness:

o Consider relocating other relevant tables (e.g., search strategies, data extraction frameworks) from the appendices to the main Methods section

**Do you want your identity to be public for this peer review?** For information about this choice, including consent withdrawal, please see our Privacy Policy

Reviewer #1: **Yes: ** Thokozile Mashaah

Reviewer #2: No

---

## [Author Response · Author response to Decision Letter 1]

9 Jun 2025

Dear Dr. Godwin Banafo Akrong, 2025/06/06

Academic Editor of PLOS ONE

Thank you for your email and the opportunity to revise our manuscript. We appreciate the time and insights of the reviewers and believe that their comments have helped us to significantly improve the manuscript. We have carefully considered all of the comments and have revised the manuscript accordingly.

Below, we provide a point-by-point response to each of the editor and reviewers' comments.

I encourage you to address all the reviewers' (Reviewer #1 and #2) comments and make the necessary revisions, particularly in improving the PRISMA flow diagram and the critical discussion of gaps and methods sections. I look forward to reviewing your revised manuscript.

Response: We acknowledge the importance of addressing all the comments provided by both Reviewer #1 and Reviewer #2. We are committed to carefully considering each point and making the necessary revisions to improve the quality and clarity of our protocol.

Journal Requirements:

Comment 1. Please ensure that your manuscript meets PLOS ONE's style requirements, including those for file naming. The PLOS ONE style templates can be found at https://journals.plos.org/plosone/s/file?id=wjVg/PLOSOne_formatting_sample_main_body.pdf and https://journals.plos.org/plosone/s/file?id=ba62/PLOSOne_formatting_sample_title_authors_affiliations.pdf.

Response: Thank you for this important point. We have carefully reviewed the PLOS ONE style guidelines and made the necessary revisions to ensure compliance. As these revisions primarily involved formatting adjustments, such as file naming and layout, we have not highlighted them. However, we confirm that all changes have been made according to the journal's requirements.

Comment 2. In the online submission form you indicate that your data is not available for proprietary reasons and have provided a contact point for accessing this data. Please note that your current contact point is a co-author on this manuscript. According to our Data Policy, the contact point must not be an author on the manuscript and must be an institutional contact, ideally not an individual. Please revise your data statement to a non-author institutional point of contact, such as a data access or ethics committee, and send this to us via return email. Please also include contact information for the third party organization, and please include the full citation of where the data can be found.

Response: No datasets were generated or analysed during the current study. All relevant data from this study will be made available upon study completion. The sentence was corrected.

Comment 3. Please remove your figures from within your manuscript file, leaving only the individual TIFF/EPS image files, uploaded separately. These will be automatically included in the reviewers’ PDF.

Response: The figures will be removed from the manuscript file, and the individual TIFF image file uploaded separately.

Reviewer #1:

This is an interesting area and the core elements of a review protocol have been included. However I have a few comments for the authors.

Comment 1. What is your definition of unmarried women? Is it never been married? Is it those who have been married before and are currently divorced/single? Is it those who are cohabiting? As this might have a bearing on your articles

Response: For the purpose of this scoping review, "unmarried women" refers to women who have never been married. (Page 5, Line 92)

Comment 2. Why are you restricting to 1990-2025 period? You might need to justify

Response: The search covers publications from inception to May 31, 2025. (Page 5, Lines 95)

Comment 3. Line 20 is the Lead author part of the independent extractors?

Response: Currently, the two authors listed are involved in the initial stages of the protocol development. Additional collaborators will join the team for the full review process, and their specific roles, including their involvement in independent data extraction, will be clearly defined and acknowledged in the final manuscript.

Comment 4. Line 33 – You might need to write ISI in full for the first time.

Response: "ISI" be written out in full as "Institute for Scientific Information" upon its first mention in the revised manuscript. (Page 6, Line 110)

Comment 5. I think you should also mention the use of wild cards and truncations to broaden the database search

Response: Thank you for the suggestion. We include a statement in the search strategy section specifying the use of wildcards and truncation to ensure a broader and more comprehensive search across the databases. (Page 6, Lines 108-109)

Comment 6. Are you also going to including the 6th optional process by Arksey and O’Malley of consultative exercise as this might provide useful information if key stakeholders are engaged in the finalization of search strategy and grey literature identification

Response: The consultative exercise, as suggested by Arksey and O'Malley, is a valuable approach. While not explicitly listed as a separate step in our 14-step framework which is based on their methodology and incorporates improvements by Tricco and Peters, the principles of engaging with key stakeholders will be integrated into the literature search and grey literature identification stages to ensure a comprehensive and relevant review. (Page 5, Lines 101-103)

Comment 7. May you consider provision of an example of the search strategy key words table

Response: We appreciate this valuable suggestion. We have added an S2 File illustrating our search strategy keyword. (Page 5, Line 111)

Comment 8. Take note that some words such as sub-fertility, bareness, sterility and infecundity might need to be included in the search strategy

Response: We appreciate this valuable feedback. We've reviewed our search strategy and have incorporated the suggested keywords. (Page 5, Line 111)

Comment 9. Who will do the search of studies? Will it be done by one person the lead author or by the independent reviewers? Will this be done separately? Are you involving any expert in Library and Information retrieval to assist with the searching?

Response: We appreciate your questions regarding our search strategy. Initial search for studies will be conducted by S.B.. We won't be seeking assistance from an external expert in Library and Information retrieval for the search itself. However, to ensure the robustness of our search strategy, we will consult with subject-matter experts in the field when developing and refining the keywords used. (Page 6, Lines 114-117)

Comment 10. Line 135 Who is S.B? Is this the lead author being referred to? Who will contact him/her?

Response: Thanks for pointing this out. We've clarified in the manuscript that S.B. refers to the principal investigator. S.B. will be contacted for any essential missing information to ensure a complete data synthesis. (Page 6, Line 114)

Comment 11. Line 141 Joanna Briggs (JBI) have developed critical appraisal tools which you might consider to use

Response: We appreciate the suggestion regarding Joanna Briggs Institute (JBI) critical appraisal tools. However, consistent with scoping review methodology, we will not be conducting a formal quality appraisal of the included studies. This approach aligns with established guidelines for scoping reviews, as cited in reference (1).

Comment 12. Line 144 You might need to include thematic analysis for the qualitative and mixed methods studies

Response: Thank you for this excellent point. We agree that thematic analysis is crucial for qualitative and mixed-methods studies. We've clarified our Data Synthesis and Analysis section to explicitly state that thematic analysis will be employed for qualitative data, in addition to the tabular and diagrammatic synthesis and narrative summary already mentioned. (Page 8, Lines 153-154)

Comment 13. Line 154 I would propose that you rephrase that since quality assessment is optional, the authors will not consider doing that process.

Response: Thank you for this crucial feedback. We removed this sentence.

Comment 14. Line 156 This is debatable on the usefulness of the finding for policy if the quality of studies are not assessed/ critically appraised. You might need to consider revising to include critical appraisal

Response: We appreciate the reviewer's point regarding the importance of critical appraisal. While we acknowledge its value, a formal quality assessment of individual studies was beyond the scope of this foundational study, which aims to provide an initial critical foundation.

Comment 15. References – You might need to consider citing articles that are more current e.g 5 years since there is quite a number of literature on infertility. The methodological citations are ok.

Response: We thank the reviewer for their valuable feedback on the currency of the references. While we appreciate the suggestion to limit citations to the last five years to reflect the most recent literature in the rapidly evolving field of infertility, we believe that a 10-year timeframe is more appropriate for this manuscript. This extended period allows for the inclusion of foundational studies and established methodological papers that provide essential context and are still widely accepted. Our approach was to balance the inclusion of seminal works with current findings to offer a comprehensive overview. We have, however, re-examined our references to ensure that the most critical recent literature is well-represented while retaining the key publications from the last decade that are fundamental to the arguments presented.

Reviewer #2:

Thank you for submitting your paper on this important and timely topic. Below are my suggestions to strengthen the manuscript:

Critical Discussion of Gaps and Methods:

Comment 1. It would be valuable to include a critical discussion of the identified gaps in male infertility screening (e.g., limitations in current guidelines, underrepresented populations, or unmet needs in screening protocols).

Response: Thank you for your valuable feedback. We note that the comment refers to male infertility screening, whereas our study focuses on infertility screening in unmarried women. In the final manuscript, we will include a critical discussion. (Page 4, Line 71)

Comment 2. Additionally, a summary of the methods currently used for infertility screening would provide useful context for readers.

Response: Thank you for the suggestion. We will include a concise summary of current infertility screening methods to provide better context for readers. (Page 3, Lines 57-62)

PRISMA Flow Diagram:

Comment 3. The PRISMA diagram is currently empty, please confirm whether this is an upload error. If not, the absence of this data raises concerns about the study’s findings, as the screening process is a cornerstone of a scoping review.

Response: Thank you for your feedback. We confirm the PRISMA diagram was inadvertently omitted due to an upload error. (Page 7, Line 128)

Comment 4. I recommend moving the PRISMA diagram from the appendices to the Methods section to improve readability and allow readers to follow the screening process seamlessly. Accompanying the diagram with a brief narrative (e.g., number of records screened, excluded, and reasons for exclusion) would further enhance clarity.

Response: Thank you for the suggestion. We will move the PRISMA diagram to the Methods section. (Page 7, Line 128)

Methods Section Clarity and Completeness:

Comment 5. Consider relocating other relevant tables (e.g., search strategies, data extraction frameworks) from the appendices to the main Methods section

Response: Thank you for the suggestion. We believe retaining tables like search strategies and data extraction frameworks as Supporting Information is more appropriate, as it keeps the main Methods section concise while ensuring detailed information remains accessible for interested readers.

Response: We add a figure in this manuscript with requirements checking.

References:

1. Tricco AC, Lillie E, Zarin W, O'Brien KK, Colquhoun H, Levac D, et al. PRISMA Extension for Scoping Reviews (PRISMA-ScR): Checklist and Explanation. Annals of Internal Medicine. 2018;169(7):467-73.

---

## [Decision Letter · Decision Letter 1]

25 Jun 2025

Dear Dr. Mogharabian,

Thank you for submitting your manuscript to PLOS ONE. After careful consideration, we feel that it has merit but does not fully meet PLOS ONE’s publication criteria as it currently stands. Therefore, we invite you to submit a revised version of the manuscript that addresses the points raised during the review process.

**ACADEMIC EDITOR:**

We look forward to receiving your revised manuscript.

Kind regards,

Godwin Banafo Akrong, Ph.D.

Academic Editor

PLOS ONE

Journal Requirements:

Reviewers' comments:

Reviewer's Responses to Questions

**Comments to the Author**

1. Does the manuscript provide a valid rationale for the proposed study, with clearly identified and justified research questions?

Reviewer #1: Yes

Reviewer #2: Yes

2. Is the protocol technically sound and planned in a manner that will lead to a meaningful outcome and allow testing the stated hypotheses?

Reviewer #1: Yes

Reviewer #2: Yes

3. Is the methodology feasible and described in sufficient detail to allow the work to be replicable?

Reviewer #1: Yes

Reviewer #2: Yes

4. Have the authors described where all data underlying the findings will be made available when the study is complete?

Reviewer #1: Yes

Reviewer #2: Yes

5. Is the manuscript presented in an intelligible fashion and written in standard English?

Reviewer #1: Yes

Reviewer #2: Yes

You may also provide optional suggestions and comments to authors that they might find helpful in planning their study.

Reviewer #1: The authors have throughly addressed the comments raised in the initial review and I recommend for submission

Reviewer #2: 1. The abstract is poorly written. The authors should please follow the criteria presented by the journal when it comes to drafting an abstract. Kindly merge the sections presented into one paragraph.

2. The inclusion and exclusion criteria should be moved to the method section of their paper.

3. The author should kindly pay attention to the tense in which they report their study since this work has already been carried out. This concern runs throughout the entire work. Example: “We won't be seeking assistance from an external expert in Library and Information retrieval for the search itself. However, to ensure the robustness of our search strategy, we will consult with subject-matter experts in the field when developing and refining the keywords used.”

**Do you want your identity to be public for this peer review?** For information about this choice, including consent withdrawal, please see our Privacy Policy

Reviewer #1: **Yes: ** Thokozile Mashaah

Reviewer #2: No

---

## [Author Response · Author response to Decision Letter 2]

25 Jun 2025

Dear Dr. Godwin Banafo Akrong, 2025/06/25

Academic Editor of PLOS ONE

Thank you for your email and the opportunity to revise our manuscript. We appreciate the time and insights of the reviewers and believe that their comments have helped us to significantly improve the manuscript. We have carefully considered all of the comments and have revised the manuscript accordingly.

Below, we provide a point-by-point response to each of the editor and reviewers' comments.

ACADEMIC EDITOR:

I encourage you to address all the concerns raised by Reviewer #2 and make the necessary revisions. I look forward to reviewing your revised manuscript.

Response: We appreciate your feedback and the opportunity to revise our manuscript. We will carefully address all the concerns raised by Reviewer #2 and make the necessary revisions to the manuscript. We are committed to submitting a revised version that fully addresses these points.

Journal Requirements:

Comment 1. Please review your reference list to ensure that it is complete and correct. If you have cited papers that have been retracted, please include the rationale for doing so in the manuscript text, or remove these references and replace them with relevant current references. Any changes to the reference list should be mentioned in the rebuttal letter that accompanies your revised manuscript. If you need to cite a retracted article, indicate the article’s retracted status in the References list and also include a citation and full reference for the retraction notice.

Response: We have thoroughly reviewed our reference list to ensure its completeness and accuracy. We confirm that none of the cited papers have been retracted. All references are current and relevant to our manuscript.

Reviewer #1:

The authors have throughly addressed the comments raised in the initial review and I recommend for submission

Response: We sincerely thank Reviewer #1 for their positive feedback.

Reviewer #2:

Comment 1. The abstract is poorly written. The authors should please follow the criteria presented by the journal when it comes to drafting an abstract. Kindly merge the sections presented into one paragraph.

Response: Thanks for your feedback on the abstract's structure. While we appreciate the suggestion to merge it into a single paragraph, we've intentionally kept the current segmented format with distinct headings. As a scoping review protocol, this detailed, sectioned approach significantly enhances clarity and navigability. It helps readers and reviewers quickly understand specific planned steps. We believe this structure best serves the purpose of a protocol, optimizing its utility. We also note that the journal guidelines do not mandate a single-paragraph format for the abstract. We've ensured the abstract adheres to all other journal guidelines, such as word count and exclusion of citations.

Comment 2. The inclusion and exclusion criteria should be moved to the method section of their paper.

Response: Thank you for your valuable comment. For a scoping review protocol, we've intentionally kept "Inclusion and exclusion criteria" as a distinct section, separate from "Methods." In a protocol, these criteria are foundational; they define the review's scope immediately after the research questions. This upfront placement enhances clarity and transparency, helping readers quickly understand the review's boundaries before delving into the specific methodological steps. While they inform the methods, they are crucial, standalone components for a protocol.

Comment 3. The author should kindly pay attention to the tense in which they report their study since this work has already been carried out. This concern runs throughout the entire work. Example: “We won't be seeking assistance from an external expert in Library and Information retrieval for the search itself. However, to ensure the robustness of our search strategy, we will consult with subject-matter experts in the field when developing and refining the keywords used.”.

Response: Thank you for your valuable feedback regarding tense usage throughout the manuscript. We appreciate you highlighting this concern. However, we would like to clarify that this manuscript is a scoping review protocol, not a completed study. As such, it outlines the planned methodology and future actions for a review that is yet to be conducted. However we revise "Won't" to "will not".

Response: We add a figure in this manuscript with requirements checking.

---

## [Decision Letter · Decision Letter 2]

21 Jul 2025

Infertility Screening in Unmarried Women: A Scoping Review Protocol

PONE-D-25-15839R2

Dear Dr. Mogharabian,

We’re pleased to inform you that your manuscript has been judged scientifically suitable for publication and will be formally accepted for publication once it meets all outstanding technical requirements.

Kind regards,

Godwin Banafo Akrong, Ph.D.

Academic Editor

PLOS ONE

Additional Editor Comments (optional):

Reviewers' comments:

Reviewer's Responses to Questions

**Comments to the Author**

1. Does the manuscript provide a valid rationale for the proposed study, with clearly identified and justified research questions?

Reviewer #2: Yes

2. Is the protocol technically sound and planned in a manner that will lead to a meaningful outcome and allow testing the stated hypotheses?

Reviewer #2: Yes

3. Is the methodology feasible and described in sufficient detail to allow the work to be replicable?

Reviewer #2: Yes

4. Have the authors described where all data underlying the findings will be made available when the study is complete?

Reviewer #2: Yes

5. Is the manuscript presented in an intelligible fashion and written in standard English?

Reviewer #2: Yes

You may also provide optional suggestions and comments to authors that they might find helpful in planning their study.

Reviewer #2: The authors have provided comprehensive responses to all the reviewers' comments and have made appropriate revisions to the manuscript.

**Do you want your identity to be public for this peer review?** For information about this choice, including consent withdrawal, please see our Privacy Policy

Reviewer #2: **Yes: ** Neema Landey

---

## [Editor Report · Acceptance letter]

PONE-D-25-15839R2

PLOS ONE

Dear Dr. Mogharabian,

I'm pleased to inform you that your manuscript has been deemed suitable for publication in PLOS ONE. Congratulations! Your manuscript is now being handed over to our production team.

Kind regards,

on behalf of

Dr. Godwin Banafo Akrong

Academic Editor

PLOS ONE